# Dyslipidemia and associated risk factors among HIV/AIDS patients on HAART in Asmara, Eritrea

Oliver Okoth Achila[1], Feven Abrhaley[2,3], Yafet Kesete[2,4]*, Feven Tesfaldet[2,5], Filmon Alazar[2,6], Lidya Fisshaye[2,7], Lidya Gebremeskel[2,8], Rodas Mehari[2,9], Danait Andemichael[2]

1 Orotta College of Medicine and Health Sciences, Asmara, Eritrea, 2 Asmara College of Health Sciences, Asmara, Eritrea, 3 National Health Laboratory, Asmara, Eritrea, 4 Nakfa Hospital, Nakfa, Eritrea, 5 Afabet Hospital, Afabet, Eritrea, 6 Ghindae Zonal Referral Hospital, Ghindae, Eritrea, 7 Halibet National Referral Hospital, Asmara, Eritrea, 8 Keren Zonal Referral Hospital, Keren, Eritrea, 9 Agordot Hospital, Agordot, Eritrea

* yafuyafa@gmail.com

**Editor:** Arturo Cesaro, University of Campania Luigi Vanvitelli Department of Translational Medicine: Universita degli Studi della Campania Luigi Vanvitelli Dipartimento di Scienze Mediche Traslazionali, ITALY

## Abstract

### Background

Though the initiation of Highly Active Antiretroviral Therapy (HAART) has led to decreased HIV/AIDS related mortality, the regimen has been reported to be associated with lipid toxicities. Baseline data on such disturbances are required to induce countrywide interventional HIV/AIDS programs. The aim of this study was to determine the frequency and risks of dyslipidemia in HIV patients on HAART medication in Eritrea.

### Methods

A cross sectional study was conducted on HIV/AIDS patients in two national referral hospitals in Asmara, Eritrea. A structured questionnaire was used to collect demographic data and blood sample was taken for analyses of lipid profile tests. Data was analyzed using chi-square test, Post Hoc and logistic regression in SPSS software.

### Results

The study included 382 participants of whom 256(67%) were females. Their median age, CD4+ T cell count (cell/microliter) and duration of HAART (years) was 45(IQR: 38–51), 434 (IQR: 294–583) & 5(IQR: 3–5) respectively. The prevalence of dyslipidemia was 331 (86.6%). Increased Low Density Lipoprotein-C (LDL-C) 213(55.8%) was the predominant abnormality. Abacavir was significantly related with highest means of triglycerides (TG) (228.17 ± 193.81) and lowest means of High Density Lipoprotein (HDL-C) (46.94 ± 12.02). Females had substantially higher proportions of TG (aOR = 2.89, 95% CI: 1.65–5.05) and TC/HDL ratio (aOR = 2.33, 95% CI: 1.40–3.87) and low HDL-C (aOR = 2.16, 95% CI: 1.34–3.48). Increased age was related with increased pro-atherogenic lipid parameters. High LDL-C was more infrequent in non-smokers (aOR = 0.028, 95% CI: 0.12–0.69).

**Data Availability Statement:** All relevant data are within the paper and its Supporting information files.

**Funding:** The author(s) received no specific funding for this work.

**Competing interests:** The authors have declared that no competing interests exist.

## Conclusion

The study showed a high prevalence of dyslipidemia in HIV-patients receiving HAART in Eritrea. Sex, age and smoking practice were among key factors associated with dyslipidemia. The necessity to assess lipid profiles and other cardiovascular risk factors before initiation of HAART treatment and continuous monitoring during therapy is mandatory.

## Introduction

Incidence of human immunodeficiency virus (HIV) infection has grown to pandemic proportions soon after early cases of acquired immunodeficiency syndrome (AIDS) were reported in 1981. Sub-Saharan Africa (SSA) is the most affected region, with an estimated over 25.6 million people living with HIV in 2020. Over two third of the new HIV infections in 2020 occurred in Sub-Saharan Africa. The subsequent introduction of HAART has changed HIV infection into a chronic manageable infectious disease. By June 2020, globally more than 21 million HIV-positive people were receiving HAART, up from 18.2 million in June 2016 and 15.8 million in 2015 [1]. This has led to a marked reduction in morbidity and mortality caused by the virus and improved quality of life of treated individuals [2].

These HAART regimens typically include a combination of at least three drugs, such as different association of protease inhibitors (PI), non-nucleoside reverse transcriptase inhibitors (NNRTI) and nucleoside reverse transcriptase inhibitors (NRTI) [3]. Customarily, HAART combines two NRTIs with either a NNRTI, or a PI [4]. The decision on which drug combination or NRTI 'backbone' to employ, and which agents to add, depends on many factors including HIV viral load, CD4+ T-cell count, drug interactions, potential toxicities, viral resistance and pill burden [5].

In SSA, the epicenter of HIV pandemic, the widely used first-line antiretroviral regimens combine two NRTI with a NNRTI. PIs are the compulsory components of the second-line treatment subsequent to the failure of the first- line therapy in which a change from efavirenz (TDF/3TC/EFV) to lopinavir (TDF/3TC/LPV/r) takes place [6]. In general, the continual development of new drugs has enabled effective suppression of viral replication and restoration of the immune system in HIV patients, which has brought a new perspective to the treatment of a disease that had previously displayed a rapid and fatal course [7].

However, early concerns of marked rates of myocardial infarction emerging as a result of dyslipidemia in HAART experienced HIV patients [8] have been noted by several researches. The D:A:D study, a large, multi-cohort prospective research, showed associations between increased myocardial infarction and exposure to antiretroviral therapy [9]. These lipid disorders often include elevations in total cholesterol (TC) and triglycerides (TG) and reduced levels of high-density lipoprotein (HDL-C) [10]. Presence of dyslipidemia then causes heart disease, heart attack and stroke by increasing the risk of blood clots which have a strong impact on mortality rates.

Shortly after the introduction of HAART medication, a syndrome of central adiposity, subcutaneous lipoatrophy and dyslipidemia, known as HIV-associated lipodystrophy (HIVLD) is observed [11]. These lipodystrophic body changes can jeopardize the quality of life of these patients, leading to low adherence to HAART and subsequent virologic and clinical failure [12]. Moreover, it is well-established that cardiovascular risk factors are more usually found in HIV-infected individuals with diabetes [13], smoking practice [14] and hypertension [15].

These changes were initially associated with exposure to protease inhibitors (PI) but subsequently exposure to nucleoside reverse transcriptase inhibitors(NRTIs) particularly stavudine (d4T) were recognized as being central to the development of this syndrome [5, 16]. Stavudine (d4T) and protease inhibitors (PIs) increase the blood levels of TC, LDL-C and TG. Nevirapine (NVP) use is associated with increases in LDL-C, whereas increases in TC and TG are observed with use of efavirenz (EFV) especially in prolonged therapy [17]. Moreover, demographic factors and disease stage can influence level of atherogenic lipid profiles [18] indicating factors unique to a region or country can influence the frequency of dyslipidemia and lipodystrophy.

To be effective, interventions aimed at reducing the dyslipidemic effects of several HAART regimen need to be based on a proper evaluation of the current therapy situation. No adequate previous studies have been conducted on the assessments of lipid health in HIV infected people in Eritrea which could have been used as reference. Therefore, the aim of this study is to evaluate lipid abnormalities and associated risk factors among HAART experienced HIV/AIDS patients in two referral hospitals in Asmara, Eritrea. This study will help to estimate the magnitude of dyslipidemia and severity of the disease and produce a baseline guide to therapy for HIV population in the respective setting.

## Methods

### Study area and design

A cross sectional study was conducted in Halibet National Referral Hospital (HNRH) and Orotta National Medical Surgical Referral Hospital (ONMSRH), in Asmara, Eritrea from March to June, 2018. They are the largest national referral hospitals with catchment area of over 814,000 inhabitants and a simultaneous huge number of referral patients visiting from different parts of the country. The study population were known HIV/AIDS positive individuals who visited ONMSRH and HNRH for routine check-up and medication.

### Sampling technique

The study was carried out with a sampling method of convenience in the sense that, participants were patients who turned up voluntarily for their routine and subsidized semester check-ups to which lipid profile tests were planned along with other laboratory monitoring tests. Study participants were HIV patients $\geq$ 18 years old and having good adherence to medication. Patients who had HAART for less than a year, receiving lipid altering therapies like statins or with known diabetes mellitus, liver disease, hepatitis B, renal failure, thyroid disease and pregnancy were excluded from the study.

### Sociodemographic data collection

A structured questionnaire was used to collect socio-demographics, medical history (diabetes mellitus, anti-dyslipidemic drug use and renal failures) and other lifestyle characteristics (alcohol consumption and smoking). Clinical history including duration since HIV diagnosis, CD4 + T cell count (cell/microliter), viral load, types and duration of HAART-regimen use at the time of blood sampling were obtained from patient records. Anthropometric measurements such as weight, height, waist and hip circumference were also measured.

### Specimen collection and analysis

After applying standard antiseptic technique, a total of 5ml of venous blood sample was obtained in a uniquely labelled chemistry tubes from each individual. Blood specimen were

then allowed to clot and serum was separated by centrifugation at 3000 rpm for 3 minutes. The serum samples were stored at 6˚C and were analyzed within 24 hours of collection for lipid parameters including TC, LDL-C, HDL-C, and TG using AU480 Chemistry Analyzer (Beckman Coulter- AU480). Low Density Lipoprotein concentration (LDL-C) was calculated by the Friedewald's formula; LDL-c (mg/dl) = TC- [HDL-c+ TG/5] [19].

In accordance with the United States National Cholesterol Education Program, Adult Treatment Panel III (NCEP-ATP III) guidelines, abnormal lipid profile was stated as TG ≥ 150 mg/dl, LDL-c ≥ 130 mg/dl, HDL-c < 40 mg/dl, TC ≥ 200 mg/dl and TC/HDL-c ratio ≥ 5 [20]. Dyslipidemia was defined as abnormal levels of the above mentioned lipid parameters of either one or more than one of those values.

## Statistical analysis

Generated data was subjected to statistical analysis using computer software (SPSS version 20.0, SPSS Inc. Chicago, IL, USA). Responses in the questionnaires and laboratory results were tabulated, coded and processed. Cross tabulations were used to analyze relationship between dependent and independent variables. Descriptive statistics was used to give clear picture of background variables using frequency distribution tables and percentages. Depending on the nature of the variables, Pearson Chi square ($\chi^2$) test/ or Fishers exact test was conducted to evaluate the relationship between independent and dependent variables. Multivariate and univariate logistic regression models were developed to establish the relationship between specific lipid profiles and associated risk factors. ANOVA and Post Hoc tests were used to compare means of the key parameters. At 95% level of significance, observed differences was considered to be significant for p<0.05.

## Quality control

Validity and content of the questionnaire was maintained through the supervision of experts in the field of laboratory medicine and infectious disease. Data and sample collectors were senior year clinical laboratory science students which were trained to ascertain a common understanding by employing in-house practice programs using role play interviews and thorough discussion sessions. For laboratory chemical tests, all the chemistry analytical equipment's were periodically undergoing calibration and quality control according to laboratory protocols prior to sample processing.

## Ethical consideration

Approval for the study was sought from the Asmara College of Health Science research ethical committee and Ministry of Health. Moreover, a written and verbal consent was obtained from study participants upon the acquisition of the data. The questionnaire contained a code for patient identification which was also used to label the blood sample to match the questionnaires. A written consent was also obtained while collecting blood sample for chemical analysis. Participants were also informed about their right to leave the study any time with no resultant consequence and standard care and respect was accorded to the targeted respondents whether they have consented or declined to participate in the study.

## Results

### Sociodemographic and clinical data

A total of 382 HIV/AIDS patients on HAART were enrolled in this study from whom 256 (67%) were females. The mean (± standard deviation [SD]) age of patients was 44.89(±10.37)

ranging from 18 to 82. The mean BMI value of patients was 21.25(±4.17). History of cardiovascular disease (CVD) and hypertension were present in 4.5% and 2.6% of the participants respectively.

All patients have received a triple-drug regimen including an NNRTI and 2 NRTIs. Patients were either on zidovudine (AZT), tenofovir (TDF) or abacavir (ABC) based drug combinations with lamivudine (3TC) regularly present in all the first-line combination regimens. The mean duration of HAART therapy at the time of blood sampling was 4.57(±2.43) years. Patient's history from their card showed that only 88 patients had a report on previous drug side effect. Lipoatrophy was reported for 36(41.86%) patients while 13(15.12%) patients had anemia as side effect of the drugs they were taking (Table 1).

**Table 1. Prevalence of dyslipidemia in HIV infected patients in Asmara, Eritrea.**

| Variables | Category | Total | Dyslipidemia | | p-value |
|---|---|---|---|---|---|
| | | | Yes n(%) | No n(%) | |
| **Gender** | Male | 126(33) | 101(80.2) | 25(19.8) | **0.009** |
| | Female | 256(67) | 230(89.9) | 26(10.2) | |
| **Age** | 18–35 | 69(18.1) | 59(85.5) | 10(14.5) | 0.272 |
| | 36–45 | 149(39) | 132(88.6) | 17(11.4) | |
| | 46–55 | 107(28) | 95(88.8) | 12(11.2) | |
| | >55 | 57(14.9) | 45(78.9) | 12(21.1) | |
| **Education** | Illiterate | 34(8.9) | 34(100) | 0(0) | 0.203 |
| | Elementary | 64(16.8) | 55(85.9) | 9(14.1) | |
| | Junior | 100(26.2) | 86(86) | 14(14) | |
| | Secondary | 154(40.3) | 131(85.1) | 23(14.9) | |
| | College | 30(7.9) | 25(83.3) | 5(16.7) | |
| **BMI** | Under weight | 91(24.1) | 68(74.7) | 23(25.3) | **0.001** |
| | Normal Weight | 225(59.7) | 205(91.1) | 20(8.9) | |
| | Over weight | 49(13.0) | 43(87.8) | 6(12.2) | |
| | Obese | 12(3.2) | 12(100) | 0(0) | |
| **Waist to hip Ratio** | Normal | 165(43.2) | 134(81.2) | 31(18.8) | **0.006** |
| | Central Obesity | 217(56.8) | 197(90.8) | 20(9.2) | |
| **CD4** | >500 | 145(38) | 124(85.5) | 21(14.5) | 0.127 |
| | 200–500 | 184(48.2) | 165(89.7) | 19(10.3) | |
| | <200 | | 42(79.2) | 11(20.8) | |
| **HAART Combination** | AZT BDC | 126(33) | 108(83.3) | 21(16.7) | 0.408 |
| | ABC BDC | 18(4.7) | 16(88.9) | 2(11.1) | |
| | TDF BDC | 238(62.3) | 210(88.2) | 28(11.8) | |
| **Duration of Current Drug** | 1–3 Years | 129(33.8) | 115(89.1) | 14(10.9) | 0.579 |
| | 3–5 Years | 161(42.1) | 138(85.7) | 23(14.3) | |
| | >5 Years | 92(24.1) | 78(84.8) | 14(15.2) | |
| **No. of Drugs used** | 1 drug | 146(38.2) | 120(82.2) | 26(17.8) | 0.176 |
| | 2 drugs | 134(35.1) | 120(89.9) | 14(10.4) | |
| | 3 drugs | 94(24.6) | 83(88.3) | 11(11.7) | |
| | 4 drugs | 8(2.1) | 8(100) | 0(0) | |

Abbreviations: AZT = zidovudine; ABC = abacavir; TDF = tenofovir; BDC = Based Drug Combination; BMI = Body Mass Index.

## Pattern of dyslipidemia

Among the HIV/AIDS patients who participated in our study, 331(86.6%) had dyslipidemia in which 230 (69.5%) were females and 101 (30.5%) were males. In terms of individual lipid markers, the order was as follows: LDL-C (55.8%), low HDL-C (44%), TG (33.2%) and high TC (29.1%). The mean ± SD levels of TG, TC, HDL-C, LDL-C, non-HDL-C, TG/HDL-C and TC/HDL C in mg/dL were 150.73 ± 112.72, 215.91 ± 45.68, 49.48 ± 12.75, 137.42 ± 39.99, 166.42 ± 43.12, 3.43 ± 3.26, and 4.55 ± 1.17, respectively. Dyslipidemia were more prevalent in females (p-value = 0.009) and 14.5% of the participants with abnormal BMI had at least one lipid abnormality (p-value<0.001). 56.8% of the dyslipidemic participants were centrally obese which was statistically significant. In a male vs. female comparison, TG values (in mg/dL) were significantly higher in men (127.85 ± 78.94 vs. 192:16 ± 39.0, p-value < 0.039). The coexistence of LDL-C + high TC was the most common type of mixed dyslipidemia in the study. TC+TG +LDL-C was the most frequent type (14.4%) in participants with abnormality in three lipid markers (see Table 2).

## Subjects' lipid profiles as per the adult treatment panel III (ATP III) risk

NCEP-ATP III guidelines were used to determine the risk of atherosclerotic cardiovascular disorder risk (ASCVD) by evaluating lipid profile of the study participants. Based on this categorical scheme, 29.1%, 18.3%, 27.5%, and 39% were in high or very high risk categories for TC, TG, LDL-C and HDL-C, respectively. A significant difference between males and females was also observed in TG and HDL-C parameters (Table 3). Similarly, TC/HDL ratio of the respondents was calculated and 127(33.2%) were at risk.

**Table 2. Frequency of dyslipidemia and isolated and mixed dyslipidemia in HIV infected patients in Asmara, Eritrea.**

| Lipid Abnormality | Female N(%) | Male N(%) | Difference(%) | Total N(%) |
|---|---|---|---|---|
| **No Lipid abnormality** | 26(51) | 25(49) | 2 | 51(13.4) |
| **Isolated dyslipidemias** | | | | |
| **One abnormality** | | | | |
| TC | 77(69.4) | 34(30.6) | 38.8 | 111(29.1) |
| TG | 61(48.0) | 66(52) | 4 | 127(33.2) |
| HDL-c | 129(76.8) | 39(23.2) | 53.6 | 168(44) |
| LDL-c | 147(69) | 66(31) | 38 | 213(55.8) |
| **Mixed Dyslipidemia** | | | | |
| **Two abnormalities** | | | | |
| TG+low-HDL-C | 44(63.8) | 25(36.2) | 27.6 | 69(18.1) |
| LDL+low-HDL-C | 61(81.3) | 14(18.7) | 62.6 | 75(19.6) |
| TC+TG | 33(57.9) | 24(42.1) | 15.8 | 57(14.9) |
| TC+LDL-C | 76(69.7) | 33(30.3) | 39.4 | 109(28.5) |
| **Three abnormalities** | | | | |
| TG+TC+HDL-C | 19(86.4) | 3(13.6) | 72.8 | 22(5.8) |
| TC+TG+LDL | 32(58.2) | 23(41.8) | 16.4 | 55(14.4) |
| **Four abnormalities** | | | | |
| TG+TC+HDL-C+LDL-C | 18(85.7) | 3(14.3) | 71.4 | 21(5.5) |
| **Dyslipidemia** | 230(69.5) | 101(30.5) | 39 | 331(86.6) |

Abbreviations: LDL = low-density lipoproteins; HDL-C = high-density lipoproteins; TG = triglycerides; TC = total cholesterol.

**Table 3. NCEP-ATP III based characterization of lipid disorders in HIV infected patients in Asmara, Eritrea.**

| NCEP-ATP III Classification | Sex | | chi-square, p-value, | Percentage(%) |
|---|---|---|---|---|
| | Female N(%) | Male N(%) | | |
| **Total cholesterol (mg/dL)** | | | | |
| Optimal (< 200) | 93(65.5) | 49(34.5) | 0.434, 0.805 | 142(37.2) |
| Borderline (200–239 | 86(66.7) | 43(33.3) | | 129(33.8) |
| High-risk (≥240) | 77(69.4) | 34(30.6) | | 111(29.1) |
| **Triglyceride (mg/dL)** | | | | |
| Normal (<150) | 195(76.5) | 60(23.5) | 32.91, **0.000** | 255(66.8) |
| Borderline high (150–199) | 31(54.4) | 26(45.6) | | 57(14.9) |
| High (≥200) | 30(42.9) | 40(57.1) | | 70(18.3) |
| **LDL-C (mg/dL)** | | | | |
| Optimal (<100) | 38(56.7) | 29(43.3) | 5.38, 0.250 | 67(17.5) |
| Near-optimal (100–129) | 69(70.4) | 29(29.6) | | 98(25.7) |
| Borderline high (130–159) | 77(68.8) | 35(31.2) | | 112(29.3) |
| High (160–189) | 48(72.7) | 18(27.3) | | 66(17.3) |
| Very high (≥190) | 24(61.5) | 15(38.5) | | 39(10.2) |
| **HDL-C** | | | | |
| Optimal (≥60) | 61(84.7) | 11(15.3) | 23.59, **0.000** | 72(18.8) |
| Borderline men (40–59) & women (50–59) | 115(71.4) | 46(28.6) | | 161(42.1) |
| High risk (<40 men) and (<50 women) | 80(53.7) | 69(46.3) | | 149(39) |

Abbreviations: LDL = low-density lipoproteins; HDL-C = high-density lipoproteins.

## Association of the different drug combinations and lipid profiles

The relationship between different drug combinations and specific lipid profiles was investigated by comparing the means of lipid parameters across the drug regimen (Table 4). Generally, participants in this study were on different HAART combinations: 238(62.3%) patients were on tenofovir based drug combination (BDC), 126(33%) patients were on zidovudine BDC, and, 18(4.7%) patients were on abacavir BDC.

In spite of small number of participants who were taking ABC containing regimens, higher proportion of these patients, i.e. (61.1%) and (66.7%) were diagnosed with increased TG level and TC/HDL ratio respectively. The means of non HDL-C and TC/HDL ratio were determined as ABC>TDF>AZT in descending order. Similarly, the mean of TG/HDL ratio of abacavir based regimens were significantly greater than zidovudine based regimens (Fig 1). However, CD4 count, BMI and waist circumference were not statistically different among the various HAART combinations.

## Factors associated with abnormal lipid profiles in HIV/AIDS patients on HAART

The association between specific lipid profiles and other independent variables like socio-demographic, anthropometric and other clinical parameters is demonstrated in Table 5. Adjusted regression models have been constructed for TC, TG, LDL, HDL and TC/HDL ratio and the presence of at least one dyslipidemia has been stratified by specific covariates (Table 6). Compared to males, females were present with higher proportions of TG (aOR = 2.89, 95% CI:1.65–5.05) and TC/HDL ratio (aOR:2.33, 95% CI:1.40–3.87) and lower proportion of HDL-C (aOR:2.16, 95% CI:1.34–3.48). Likewise, the level of all lipid parameters has increased with the increase in participant's age.

**Table 4. Mean and SD of the lipid profile among three HAART drug combinations.**

| Variable | Drug Combination | Mean ± SD | 95% CI | P-Value | Post Hoc(Tukey) |
|---|---|---|---|---|---|
| **TC** | AZT BDC | 213.59±47.96 | 205.13-222.04 | 0.053 | |
| | ABC BDC | 241.11±48.26 | 217.11-265.11 | | |
| | TDF BDC | 215.24±43.84 | 209.64-220.83 | | |
| **TG** | AZT BDC | 139.85 ± 96.612 | 122.82-156.88 | **0.008** | ABC>TDF >AZT |
| | ABC BDC | 228.17 ± 193.811 | 131.79-324.55 | | |
| | TDF BDC | 150.64 ± 110.870 | 136.48-164.80 | | |
| **LDL** | AZT BDC | 133.33 ± 40.104 | 126.25- 140.4 | 0.228 | |
| | ABC BDC | 148.50 ± 33.45 | 131.86-165.14 | | |
| | TDF BDC | 138.75 ± 40.28 | 133.60- 143.89 | | |
| **HDL-c** | AZT BDC | 52.90±15.15 | 50.23-55.58 | **0.001** | AZT >TDF |
| | ABC BDC | 46.94±12.02 | 40.97-52.92 | | |
| | TDF BDC | 47.87±10.99 | 46.46- 49.27 | | |
| **TG/HDL Ratio** | AZT BDC | 3.08±2.82 | 2.58-3.58 | **0.040** | ABC > AZT |
| | ABC BDC | 5.13±3.99 | 3.15-7.12 | | |
| | TDF BDC | 3.49±3.38 | 3.06-3.93 | | |
| **TC/HDL ratio** | AZT BDC | 4.25±1.15 | 4.05-4.45 | **<0.001** | ABC>TDF> AZT |
| | ABC BDC | 5.32±1.19 | 4.73-5.91 | | |
| | TDF BDC | 4.65±1.15 | 4.50-4.79 | | |
| **LDL/HDL Ratio** | AZT BDC | 2.64±0.86 | 2.49-2.80 | **0.001** | ABC>TDF> AZT |
| | ABC BDC | 3.29±0.85 | 2.86-3.72 | | |
| | TDF BDC | 2.99±0.95 | 2.87-3.11 | | |
| **Non-HDL** | AZT BDC | 160.68±43.98 | 152.92-168.43 | **0.007** | ABC>TDF> AZT |
| | ABC BDC | 194.16±44.69 | 171.94-216.39 | | |
| | TDF BDC | 167.36±41.82 | 162.02-172.71 | | |
| **Waist Circumference** | AZT BDC | 80.21± 11.90 | 78.12-82.31 | 0.788 | |
| | ABC BDC | 82.22±10.45 | 77.02-87.42 | | |
| | TDF BDC | 80.51±11.45 | 79.05-81.97 | | |
| **BMI** | AZT BDC | 21.55± 4.27 | 20.80-22.31 | 0.324 | |
| | ABC BDC | 22.15± 4.34 | 19.99-24.32 | | |
| | TDF BDC | 21.02±4.10 | 20.49-21.54 | | |
| **CD4 Count** | AZT BDC | 479.02±202.40 | 443.34-514.7 | 0.119 | |
| | ABC BDC | 390.91±189.51 | 296.67-485.16 | | |
| | TDF BDC | 437.37±230.25 | 407.96-466.77 | | |

Abbreviations: AZT = Zidovudine; ABC = Abacavir; TDF = Tenofovir; BDC = Based Drug Combination; BMI = Body Mass Index; LDL = low-density lipoproteins; HDL-C = high-density lipoproteins; TG = triglycerides; TC = total cholesterol.

Overweight and obese participants had a notable increase in TC, TG and TC/HDL ratios. Increased TC were significantly associated with subjects who had a previous history of CVD (cOR = 3.78, 95% CI: 1.38–10.07) and hypertension (cOR = 6.01, 95% CI: 1.52–23.6). High LDL-C was more infrequent in non-smokers (aOR = 0.028, 95% CI: 0.12–0.69, p-value<0.0). Furthermore, different types of HAART regimen were significantly associated with increased TG (ABC (aOR = 4.42, 95% CI: 1.20–14.9)), High HDL (1D (aOR = 3.13, 95% CI: 1.41–6.94), TDF (aOR = 2.26, 95% CI: 1.29–3.95), ABC (aOR = 3.78, 95% CI: 1.25–11.4) and High TC/ HDL ratios (1D (aOR = 2.56, 95% CI: 1.01–6.48), TDF (aOR = 2.32, 95% CI: 1.17–4.60), ABC (aOR = 8.01, 95% CI: 2.25–28.5). Lastly, increased duration of HAART use was also noted to be related with poor lipid profiles.

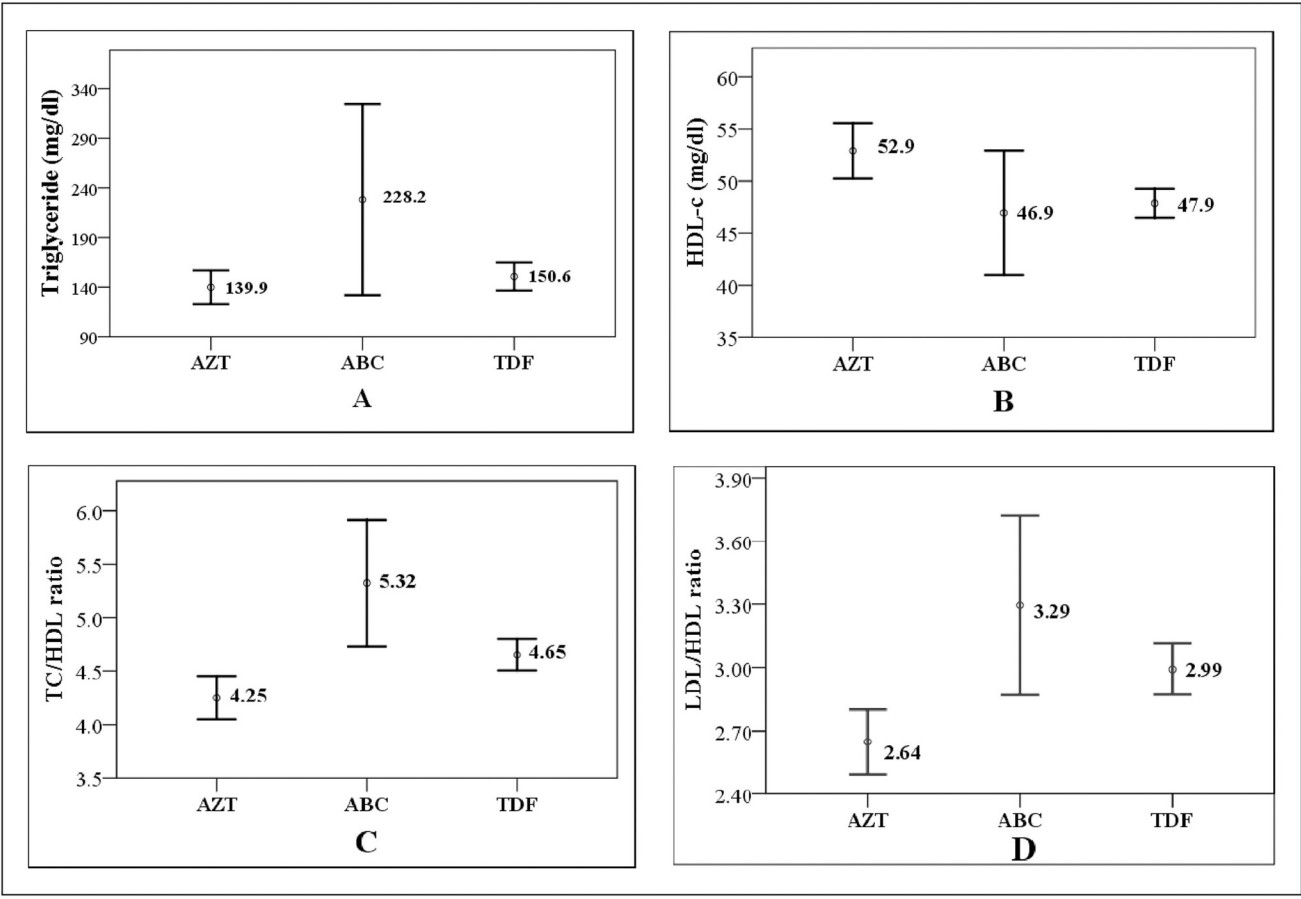

**Fig 1. A-D: Schematic representation of lipid parameters and HAART regimen.** Indicated are lipid parameters with significant differences among HAART regimen (p < 0.05). The data are exhibited as mean ± CI 95% after one-way ANOVA analysis. AZT: zidovudine; ABC: abacavir; TDF: tenofovir; CI: confidence interval.

## Discussion

This cross sectional study assessed the prevalence and characteristics of lipid profile derangements in HIV infected patients on HAART medication. The main findings include the severe burden of dyslipidemia and the importance of sex, age, anthropometric measures, smoking practice and effect of several HAART regimen on plasma lipids of HIV infected people.

The prevalence of dyslipidemia in this study was 86.6%. Though prevalence estimates of dyslipidemia may vary due to differences in definition criteria, this was one of the highest in SSA jurisdiction along with findings from Ethiopia [21] and Cameroon [22]. Unfortunately, the prevalence was also higher than what has been reported outside SSA [23].

In this study, the most prevalent type of lipid abnormality was high LDL-C (55.8%). Generally, increased means of lipid parameters were observed which coexisted with a notable disparity among males and females, high burden of hypertension and CVD. The prevalence of low HDL- C in the HIV patients was 44% on which females had more than two-fold low HDL-C than their counterparts. As high LDL-C and low HDL-C are recognized an independent risk factors for coronary artery disease, the consistent findings of both may easily grow the burden of cardiovascular diseases by undesirable amounts [24].

**Table 5. Univariate logistic regression model for factors associated with abnormal lipid profiles among HIV infected patients in Asmara, Eritrea.**

| Variables | TC>240 mg/dl COR (95% CI) | TG>200mg/dl COR(95% CI) | LDL>130mg/dl COR(95% CI) | HDL<40mg/dl COR(95% CI) | TC/HDL COR(95% CI) |
|---|---|---|---|---|---|
| **Gender** | | | | | |
| **Male** | 0.86(0.53-1.38) | **3.64(2.12-6.24)** | 0.82(0.53-1.25) | **0.44(0.28-0.69)** | **2.43(1.56-3.80)** |
| Female | Ref | Ref | Ref | Ref | Ref |
| Age | | | | | |
| 18-30 | Ref | Ref | Ref | Ref | Ref |
| 30-40 | **2.16(1.19-3.90)** | 1.27(0.65-2.45) | **2.12(1.30-3.45)** | 0.55(0.34-0.90) | **2.32(1.33-4.04)** |
| 40-50 | **2.73(1.44-5.17)** | **3.1(1.41-9.55)** | **2.61(1.42-5.10)** | 0.35(0.19-0.61) | **2.16(1.17-3.97)** |
| >50 | **2.07(1.01-10.8)** | **2.89(1.09-7.99)** | 1.81(1.04-3.12) | **0.44(0.16-0.59)** | **1.16(1.02-3.28)** |
| **BMI** | | | | | |
| <18 | Ref | Ref | Ref | Ref | Ref |
| 18-25 | **3.21(1.61-6.42)** | **6.05(2.12-17.3)** | **2.78(1.68-4.62)** | 1.14(0.69-1.85) | **2.58(1.41-4.73)** |
| 25-30 | **6.43(2.76-14.9)** | **8.70(2.68-28.3)** | **3.98(1.89-8.37)** | 1.05(0.52-2.11) | **4.14(1.91-9.03)** |
| >30 | **14.5(3.75-56.4)** | 4.35(0.71-26.8) | 3.51(0.98-12.6) | 1.95(0.58-6.62) | **9.38(2.51-34.9)** |
| **Smoking** | | | | | |
| Non-Smoker | 1.18(0.48-2.88) | **0.28(0.13-0.64)** | **0.23(0.11-0.60)** | 1.15(0.52-2.55) | 0.99(0.43-2.28) |
| Smoker | Ref | Ref | Ref | Ref | Ref |
| **Waist to Hip Ratio** | | | | | |
| Normal | Ref | Ref | Ref | Ref | Ref |
| Central Obesity | 1.52(0.96-2.39) | **1.95(1.11-3.40)** | 1.47(0.98-2.22) | 1.21(0.81-1.83) | **1.88(1.21-2.94)** |
| **Type of Drug regimen** | | | | | |
| 1C | Ref | Ref | Ref | Ref | Ref |
| 1D | 0.49(0.19-1.17) | 1.0(0.36-2.74) | 0.76(0.36-1.57) | **2.74(1.28-5.83)** | 2.13(0.93-4.89) |
| TDF | 0.86(0.50-1.48) | 1.14(0.57-2.26) | 1.27(0.77-2.11) | **1.92(1.28-3.28)** | **2.25(1.22-4.13)** |
| ABC | 2.15(0.77-6.06) | **3.38(1.1-10.3)** | 1.43(0.50-4.04) | **2.85(1.01-8.07)** | **8.25(2.68-25.3)** |
| **No. of Substituted Drug Regimen** | | | | | |
| One | Ref | Ref | Ref | Ref | Ref |
| Two | 1.45(0.85-2.47) | **3.08(1.53-6.18)** | 1.11(0.69-1.77) | 1.30(0.81-2.10) | 1.44(0.86-2.42) |
| Three | 1.54(0.86-2.75) | **3.12(1.48-6.57)** | 1.26(0.75-2.13) | 1.39(0.82-2.35) | 2.27(1.31-3.96) |
| Four | **9.88(1.91-51.2)** | **6.14(1.32-28.6)** | 6.27(0.75-52.3) | 0.91(0.21-3.96) | 4.91(1.12-21.6) |
| **Duration on ART (years)** | | | | | |
| 1-5 | Ref | Ref | Ref | Ref | Ref |
| 5-10 | **2.10(1.26-3.52)** | 1.58(0.85-2.94) | 1.01(0.64-1.57) | 0.95(0.61-1.48) | 1.41(0.87-2.28) |
| 10+ | 1.13(0.52-2.45) | 2.19(0.98-4.93) | 0.77(0.40-1.46) | 1.17(0.61-2.23) | 1.62(0.82-3.18) |

Abbreviations: 1C = Zidovudine+Lamuvidine+Neverapine; 1D = Zidovudine+Lamuvidine+Efaverenz; ABC = Abacavir; TDF: Tenofovir; LDL = low-density lipoproteins; HDL-C = high-density lipoproteins; TG = triglycerides; TC = total cholesterol.

Whilst, the major guidelines for lipid health management are based on LDL-C concentrations, several studies report that predictive values of apolipoprotein B to apolipoprotein A1 ratio and TC/HDL-C ratio level is consistently similar and provide more information about the CVD risk than a single lipoprotein measurement [25]. According to our study, 33.2% of the patients were present with abnormal TC/HDL ratio (>5) which is higher than other studies in SSA jurisdiction [22].

TC was found to be significantly higher in those patients with increased CD4 count (p-value<0.05). Though severe immune suppression (low CD4 count) has been associated with dyslipidemia in HAART-naive HIV patient [26], in a cohort of persons doing well on HAART

**Table 6. Multivariate logistic regression model for factors associated with abnormal lipid profiles among HIV infected patients in Asmara, Eritrea.**

| Variables | TC>240 mg/dl AOR (95% CI) | TG>200mg/dl AOR(95% CI) | LDL>130mg/dl AOR(95% CI) | HDL<40mg/dl AOR(95% CI) | TC/HDL AOR(95% CI) |
|---|---|---|---|---|---|
| **Gender** | | | | | |
| Male | | 2.89(1.65-5.05) | | Ref | 2.33(1.40-3.87) |
| Female | | Ref | | 2.16(1.34-3.48) | Ref |
| **Age** | | | | | |
| 18-30 | Ref | Ref | Ref | 3.23(1.49-6.99) | Ref |
| 30-40 | 3.9(1.52-10.42) | 2.32(0.95-5.65) | 2.07(1.11-3.84) | 1.79(0.92-3.49) | 2.11(0.94-4.72) |
| 40-50 | 4.8(1.7-13) | 3.7(1.46-9.35) | 2.65(1.36-5.18) | 1.41(0.69-2.85) | 2.48(1.07-5.79) |
| >50 | 3.3(1.13-9.8) | 2.98(1.12-7.93) | 1.71(0.79-3.68) | Ref | 1.27(0.49-3.28) |
| **BMI** | | | | | |
| <18 | Ref | Ref | Ref | | Ref |
| 18-25 | 2.2(1.05-4.8) | 2.98(1.43-6.20) | 2.13(1.21-3.74) | | 1.90(0.94-3.84) |
| 25-30 | 2.6(0.8-8.2) | 4.85(1.86-12.6) | 1.88(0.72-4.94) | | 2.28(0.75-6.91) |
| >30 | 6.9(1.2-39.2) | 10.6(2.29-49.3) | 1.45(0.30-6.97) | | 6.01(1.03-35.1) |
| **Smoking** | | | | | |
| Non-Smoker | | | 0.28(0.12-0.69) | | |
| Smoker | | | Ref | | |
| **Waist to Hip Ratio** | | | | | |
| Normal | | Ref | | | |
| Central Obesity | | 1.97(1.14-3.39) | | | |
| **Type of Drug regimen** | | | | | |
| 1C | | Ref | | Ref | Ref |
| 1D | | 1.16(0.42-3.15) | | 3.13(1.41-6.94) | 2.56(1.01-6.48) |
| TDF | | 1.33(0.66-2.67) | | 2.26(1.29-3.95) | 2.32(1.17-4.60) |
| ABC | | 4.42(1.20-14.9) | | 3.78(1.25-11.4) | 8.01(2.25-28.5) |
| **No. of Substituted Drug Regimen** | | | | | |
| One | Ref | Ref | | | |
| Two | 1.03(0.52-2.0) | 2.71(1.32-5.52) | | | |
| Three | 0.94(0.44-2.01) | 3.71(1.61-8.54) | | | |
| Four | 11.7(1.09-125.2) | 2.07(0.32-13.4) | | | |
| **Duration on HAART (years)** | | | | | |
| 1-5 | | 1.19(0.47-3.02) | | | |
| 5-10 | | 0.42(0.20-0.89) | | | |
| 10+ | | Ref | | | |

Abbreviations: 1C = Zidovudine+Lamuvidine+Neverapine; 1D = Zidovudine+Lamuvidine+Efaverenz; ABC = Abacavir; TDF: Tenofovir; LDL = low-density lipoproteins; HDL-C = high-density lipoproteins; TG = triglycerides; TC = total cholesterol.

with a sufficiently improved mean CD4 count, there is only little variability in the TC of the group. Therefore, the difference in dyslipidemia of the current patients cannot be attributed to the variations in CD4 count.

In this study, abacavir was found to be significantly associated with the highest means of lipid parameters among the other drug combinations. This is consistent to the recent ACTG 5202 study that compared abacavir and tenofovir based drug regimens in which abacavir use was associated with significant greater increases in median TG (25mg/dl vs 3 mg/dl) and TC (34mg/dl vs 26mg/dl) than tenofovir at 48 weeks [5].

Similar results were also observed in the previous HEAT study, which compared abacavir and tenofovir-containing HAART in treatment-naïve patients. Those on abacavir were present with greater increases in TG (64mg/dl vs 38 mg/dl) and TC (32 mg/dl vs 23mg/dl) at 48 weeks [27]. Studies evaluating in vitro and in vivo effects of abacavir on leukocyte adhesion suggest a potential mechanism that could underlie this association [25].

Generally, zidovudine based combinations were associated with more favorable lipid panel in this study. This finding is broadly similar to what has been articulated in established literature in which, compared to zidovudine, significant smaller increases in TC and LDL-C being observed with tenofovir use [28].

A head to head comparison between the NNRTI drug regimens was also attempted. In univariate analysis, neverapine was significantly related with positive lipid profile than efaverenz (cOR = 2.02, 95% CI: 1.23–3.9) and lopinavir (cOR = 8.02, 95% CI: 2.15–29.7). This is comparable to cohort studies reporting a more favorable lipid profile for nevirapine at 48 weeks upon comparison with PI containing regimens [22].

In terms of socio-demographic characteristics, our results suggest that dyslipidemia, TG and TC/HDL ratios were independently related to the sex of participants. Although this was a cross-sectional study by design, the outcomes were broadly in keeping with those of prospective cohort studies [29]. Previously, studies in lower-income countries have discovered female sex as an independent risk factor for lipodystrophy [30]. In Rwandan research of 2190 HIV-infected patients, women were reported to have 9.7 times higher risk of developing lipodystrophy [31]. This disparity is explained by women being disadvantaged to receive adequate health care, support and education for HIV/AIDS and may be more vulnerable to the metabolic consequences and social impact of stigma [32]. Moreover, HIV-related fat distribution abnormalities, commonly associated with lipid disturbances, have been described in the past as more frequent in women [33]. Low HDL-C was another parameter that had significant association with sex in this study. This relationship is common and well recognized in treatment guidelines such as NCEP-ATP which has a different threshold for men and women.

To large extent, the data relating abnormal lipid concentrations across the lifetime of HIV infected people is augmented in this study by investigation showing a strong association between age of participants and multiple lipid markers. TC, TC/HDL ratio, LDL-C and TG which are independent CVD risk factors were demonstrated to increase with age. It is well known that increased age is related to a pro-atherogenic lipid profile. In fact, several cross-sectional population studies have demonstrated that TC and LDL levels increase after the onset of puberty until 50 years of age, and then plateau until 70 years of age [34]. Vascular aging is suggested in Cox model study as core cause of impairment contributing to the higher rates of CVD observed in HIV-infected patients [35].

In this study, the duration of exposure to HAART was found to be significantly and positively associated with raised TG value. There are suggestions that the amount of lipid profile derangements induced by HAART showing variation with duration of treatment, across populations and setting. HAART is associated with a cardio-protective lipid profile in the short term because after initiation of the medication, lipid levels return to baseline levels but soon they rise above pre-seroconversion levels in the long term [36]. Current guidelines of the CDC of Eritrea do not include routine monitoring of lipid parameters for patients receiving first line HAART initially. It might be therefore reasonable to recommend that monitoring of lipid profile should be instituted shortly after the starting of the first line HAART in all HIV patients in Eritrea.

In a distinguished analysis, we established a strong positive cross-sectional association between BMI, waist circumference (WC), hip circumference (HC), waist to hip ratio (surrogate marker of visceral adiposity) and abnormalities in most of the lipid markers. A prominent

linear-by-linear association was also notable. Though the importance of waist and hip measurements is well documented and appears to play a crucial role in this setting, its concerted application is not well established in Eritrea. Despite being cheap-to-measure marker, it is rarely assessed by clinicians, and awareness regarding the risks of increased values is low. This nonetheless, the simultaneous use of BMI and waist/hip measurements should be encouraged in treatment guidelines of HIV patients in Eritrea.

Regarding the analysis of smoking and its association to disparate lipid disorders, our outcomes suggest that non-smokers were found to have related with favorable TC levels (aOR = 0.28, 95% CI: 0.12–0.69). This finding is coherent with current knowledge that, compared with non-smokers, cigarettes smokers had poorer lipid profiles both in HIV patients and even amongst the general population [37]. Even though, information bias associated with under-reporting of undesirable lifestyles can be the case here, such prominent odds ratio is unlikely to be a chance finding. Thus, smoking cessation would be vehemently recommended.

In this study, the sample represents HIV infected people mostly acquired through convenience systematic sampling. As such, it can be contended that the sample was not fully randomized. Nonetheless, this mode of data acquisition is prominent in most studies from the region. Additional methodological issues that can undermine implication of the research were the cross-sectional nature of the study and the lack of fasting plasma glucose and blood pressure measurements which would have allowed assessment of metabolic syndrome in patients.

Lastly, the significance of this kind of research is substantially magnified in countries of SSA where very little is known about the general health of the HIV infected people. Although the study is preliminary, it is our belief that it can contribute critically to a better understanding of the burden of dyslipidemias and its associated risk factors in HIV infected people in this jurisdiction.

## Conclusion

Various important findings came forth from this study. Primarily, HIV-infected patients receiving WHO-recommended HAART drug combinations had an alerting prevalence of lipid profile derangements. All the study participants were unaware of their clinical condition. The most dominant form of dyslipidemia was high LDL-C and according to NCEP-ATP III risk strata, more than quarter of the participants were in high or very high risk categories for all lipid parameters. Study participants on abacavir based drug combination were diagnosed with the highest proportion of abnormal lipid markers. Patients on zidovudine were the least affected by poor lipid health among the different HAART regimens. On the ground of the study results, it is our view that a large proportion of the participants is at danger of developing or exacerbating pre-existing CVD. Women were disproportionately affected across most of these categories. Moreover, the stepwise multivariate modeling established that the proportion of dyslipidemia was related with sex, age, BMI, waist/hip ratio, smoking, type of HAART regimen taken and duration of medication in years. The percentage of the study participants presenting with abdominal obesity, a strong surrogate marker of CVD, was also high. Taken in concert, these findings indicate the necessity to assess lipid profiles and other cardiovascular risk factors before initiation of HAART treatment and continuous examination during therapy so that any negative effects can be optimally managed.

## Supporting information

**S1 Appendix. This contains consent form and questionnaire used for data collection.** (PDF)

## Acknowledgments

Special thanks to all staff members of Orotta and Halibet hospitals and especially the VCT (Voluntary Counseling and Treatment) and pharmacy departments for their unreserved help in all data collection processes. Our appreciation also extends to the laboratory department of Sembel Hospital for their intensive cooperation in sample analysis. We thank our subjects for their willingness to participate in the study. Finally, we would like to appreciate to all the people who have contributed valuable support for conducting this research successfully.

## Author Contributions

**Conceptualization:** Oliver Okoth Achila, Feven Tesfaldet, Filmon Alazar, Lidya Fisshaye, Lidya Gebremeskel, Rodas Mehari.

**Data curation:** Feven Abrhaley, Yafet Kesete, Filmon Alazar, Lidya Fisshaye, Lidya Gebremeskel.

**Formal analysis:** Oliver Okoth Achila, Feven Abrhaley, Yafet Kesete, Lidya Fisshaye, Lidya Gebremeskel, Danait Andemichael.

**Funding acquisition:** Oliver Okoth Achila, Danait Andemichael.

**Investigation:** Feven Abrhaley, Yafet Kesete, Feven Tesfaldet, Filmon Alazar, Lidya Fisshaye, Lidya Gebremeskel, Rodas Mehari.

**Methodology:** Oliver Okoth Achila, Feven Abrhaley, Yafet Kesete, Feven Tesfaldet, Filmon Alazar, Lidya Fisshaye, Lidya Gebremeskel, Rodas Mehari, Danait Andemichael.

**Project administration:** Danait Andemichael.

**Resources:** Yafet Kesete.

**Supervision:** Danait Andemichael.

**Validation:** Oliver Okoth Achila, Feven Tesfaldet, Rodas Mehari, Danait Andemichael.

**Writing – original draft:** Feven Abrhaley, Yafet Kesete, Feven Tesfaldet.

**Writing – review & editing:** Feven Abrhaley, Yafet Kesete.

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
