## [Decision Letter · Decision Letter 0]

20 Apr 2022

PONE-D-22-02948Lipid abnormalities and associated risk factors among HIV/AIDS patients on HAART in Asmara, EritreaPLOS ONE

Dear Dr. Kesete,

Thank you for submitting your manuscript to PLOS ONE. After careful consideration, we feel that it has merit but does not fully meet PLOS ONE’s publication criteria as it currently stands. Therefore, we invite you to submit a revised version of the manuscript that addresses the points raised during the review process.

We look forward to receiving your revised manuscript.

Kind regards,

Arturo Cesaro, MD

Academic Editor

PLOS ONE

Journal Requirements:

Additional Editor Comments (if provided):

You are invited to consider the reviewers' comments, reported at the end of this letter, and to revise your manuscript accordingly. In the letter accompanying your resubmission, please explain your response to each of the comments. Please observe the word count and citation style. For further details, please consult the Instructions for Authors on the website

Reviewers' comments:

Reviewer's Responses to Questions

**Comments to the Author**

1. Is the manuscript technically sound, and do the data support the conclusions?

Reviewer #1: Yes

Reviewer #2: Partly

2. Has the statistical analysis been performed appropriately and rigorously? 

Reviewer #1: Yes

Reviewer #2: N/A

3. Have the authors made all data underlying the findings in their manuscript fully available?

Reviewer #1: Yes

Reviewer #2: Yes

4. Is the manuscript presented in an intelligible fashion and written in standard English?

Reviewer #1: Yes

Reviewer #2: No

5. Review Comments to the Author

Reviewer #1: Review comments

Title: Lipid abnormalities and associated risk factors among HIV/AIDS patients on HAART in Asmara, Eritrea

Submission ID: PONE-D-22-02948

The authors assess dyslipidemia associated with HAART in HIV/AIDS patients which an important topic to uncover the negative impacts of the treatment. So far, many studies elucidated this scenario and I appreciate the novelty of this study conducted in Eritrea for the first time. The point by point concerns are detailed below.

Title: the title better be “Dyslipidemia…..

Abstract:

Acronyms like HAART, LDL, TC, CVD, etc… should be fully written on their first appearance

In the results section, CD4 should be corrected as “CD4+ T cell count (cell/microliter)”. In addition, results with a significant association at aOR should be considered the real predictors. cOR data cannot be inferred or reported as final outcome of association

In the conclusion factors significantly associated with dyslipidemia should be highlighted

Results

In the results, the cOR values better be displayed for readers convenience

Discussion

Subtitles in the discussion could be removed. Besides, the limitations of the study should be stated at the last paragraph of the discussion. Then the importance of the study can be mentions

Conclusion

The conclusion should not contain results values like percentages. The conclusion should be a single paragraph containing the burden of dyslipidemia (not in percentages, rather saying either “significantly high” or other means of expression). Then important predictors are mentioned and lastly a brief recommendation should be included.

Reviewer #2: Dear Authors, thank you for your work. The data are interesting but are presented in an unclear way. The text is very difficult to read, especially the presentation of the results. Furthermore, there are long and repetitive sections within the methods with not important data while some fundamental aspects are totally absent, such as the timing of the collection of the lipid profile in these patients.

1) At what time of HAART therapy were the patients evaluated for lipid profile?

2) Do we have baseline or pre-therapy values in at least some of the patients?

3) Furthermore, the therapy taken by patients should be better analyzed, looking for the use of statins in at least part of them.

4) A complete revision of the English language by native speakers is extremely necessary.

5) There are also widespread grammatical errors and typos.

6. PLOS authors have the option to publish the peer review history of their article (what does this mean?). If published, this will include your full peer review and any attached files.

Reviewer #1: **Yes: **HYLEMARIAM MIHIRETIE MENGIST

Reviewer #2: No

---

## [Author Response · Author response to Decision Letter 0]

29 Apr 2022

Rebuttal Letter: Response to Reviewers

Reviewer 1

1. Abbreviation issues: Acronyms like HAART, LDL, TC, CVD, etc… should be fully written on their first appearance. DONE.

All acronyms that has been mentioned in the study are written in full in their first mention. 

2. In the results section, CD4 should be corrected as “CD4+ T cell count (cell/microliter)”.

“CD4” has now been corrected as “CD4+ T cell count (cell/microliter)” in the revised manuscript. 

3. Results with a significant association at aOR should be considered the real predictors not cOR data.

General frequencies and the key risk factors which have showed significant association in multivariate regression analysis are indicated in the abstract section. Similar approach has also implemented in the conclusion section. 

4. In the results, the cOR values better be displayed for readers convenience.

Univariate regression analysis table is now included in the manuscript. 

5. In the conclusion, factors significantly associated with dyslipidemia should be highlighted.

The main risk factors that have been revealed statistically are outlined in the conclusion section. Necessary interventional recommendation have also been included. 

6. Subtitles in the discussion could be removed.

The discussion section is now summarized under one title detailing about all the prevalence, effect of drug regimens and factors associated with dyslipidemia. 

7. The limitations of the study should be stated at the last paragraph of the discussion. Then the importance of the study can be mentions.

The last paragraph of the discussion section is rewritten according the reviewers guide. 

8. The conclusion should be a single paragraph without result values like percentages and showing burden of dyslipidemia in other means of expression. 

- The conclusion is formatted now in one paragraph showing the key findings of the study and other important predictors. A brief recommendation is also included in the last lines of the paragraph.

Reviewer 2

9. There are long and repetitive sections within the methods with not important data.

The methods part is now summarized and details on inclusion criteria, data collection have been revised to indicate the key information regarding study participants.

10. At what time of HAART therapy were the patients evaluated for lipid profile?

At the time of blood sampling, the duration of HAART use in the patients ranged from 1 year to 16 years. The mean duration of HAART use was 4.57(±2.43) years. The prevalence of dyslipidemia worsens across time after ART initiation, despite initial favorable changes in TG and HDL during the 6 months period. To minimize this apparent effect in this study, only patients who had HAART for more than one year were included. 

This has been explained in the second paragraph of the results portion.

11. Do we have baseline or pre-therapy values in at least some of the patients?

Due to the cross sectional nature of the study, lipid profile assessments were conducted during the therapy time. Currently, evaluation of fat levels in HIV patients in Eritrea is not a regular package of the pre-therapy investigations like CD4 count, viral load, kidney function and blood sugar. Some few patients may have lipid profile tests during first stages of therapy period by specific physician order to evaluate complications related to the disease. 

One of the aim of this study is to include lipid profile tests in all HIV patients immediately before the initiation of HAART treatment and to sustain its continuous examination during therapy. This have been outlined at the last paragraph of the conclusion portion. 

12. The therapy taken by patients should be better analyzed, looking for the use of statins in at least part of them.

All therapies including accessory drugs taken by patients for other complications related to the disease were assessed. Patients who were taking any lipid altering or anti-dyslipidemic drugs were primarily excluded from the study. These include statins like, but not limited to, atorvastatin, fluvastatin, lovastatin and rosuvastatin which are commonly prescribed in the study setting. Moreover, patients who were generally taking fibrates, bile acid sequestrants, niacin and ezetimibe were also excluded for their effect in altering lipid values in patients.

 This has been indicated at the sampling technique and inclusion criteria subtitles of the methodology portion. 

13. The text is very difficult to read, especially the presentation of the results. A complete revision is needed by native speakers. There are also widespread grammatical errors and typos.

The manuscript have been revised for spelling and typing errors. Moreover, the text have been overviewed by two English native speakers for grammatical issues. Portions which were not clear have been corrected in the revised manuscript.

---

## [Decision Letter · Decision Letter 1]

20 Jun 2022

Dyslipidemia and associated risk factors among HIV/AIDS patients on HAART in Asmara, Eritrea

PONE-D-22-02948R1

Dear Dr. Kesete,

We’re pleased to inform you that your manuscript has been judged scientifically suitable for publication and will be formally accepted for publication once it meets all outstanding technical requirements.

Kind regards,

Arturo Cesaro, MD

Academic Editor

PLOS ONE

Additional Editor Comments (optional):

The manuscript appears to be improved after editing in accordance with the reviewers' comments.

Reviewers' comments:

Reviewer's Responses to Questions

**Comments to the Author**

1. If the authors have adequately addressed your comments raised in a previous round of review and you feel that this manuscript is now acceptable for publication, you may indicate that here to bypass the “Comments to the Author” section, enter your conflict of interest statement in the “Confidential to Editor” section, and submit your "Accept" recommendation.

Reviewer #1: All comments have been addressed

Reviewer #2: All comments have been addressed

2. Is the manuscript technically sound, and do the data support the conclusions?

Reviewer #1: Yes

Reviewer #2: Yes

3. Has the statistical analysis been performed appropriately and rigorously? 

Reviewer #1: Yes

Reviewer #2: Yes

4. Have the authors made all data underlying the findings in their manuscript fully available?

Reviewer #1: Yes

Reviewer #2: Yes

5. Is the manuscript presented in an intelligible fashion and written in standard English?

Reviewer #1: Yes

Reviewer #2: Yes

6. Review Comments to the Author

Reviewer #1: The significantly associated factors should be clearly indicated in the table. use either asterisks or other symbols.

Reviewer #2: Dear authors,

Thanks to have replied to my comments. I think that the text has been improved after revision

7. PLOS authors have the option to publish the peer review history of their article (what does this mean?). If published, this will include your full peer review and any attached files.

Reviewer #1: **Yes: **HYLEMARIAM MIHIRETIE MENGIST

Reviewer #2: No

---

## [Editor Report · Acceptance letter]

23 Jun 2022

PONE-D-22-02948R1 

*Dyslipidemia and associated risk factors among HIV/AIDS patients on HAART in Asmara, Eritrea*

Dear Dr. Kesete:

I'm pleased to inform you that your manuscript has been deemed suitable for publication in PLOS ONE. Congratulations! Your manuscript is now with our production department. 

Kind regards, 

on behalf of

Dr. Arturo Cesaro 

Academic Editor

PLOS ONE